# Development and Validation of a Duplex RT-qPCR for Detection of Peach Latent Mosaic Viroid and Comparison of Different Nucleic-Acid-Extraction Protocols

**DOI:** 10.3390/plants12091802

**Published:** 2023-04-27

**Authors:** Marta Luigi, Anna Taglienti, Carla Libia Corrado, Marco Cardoni, Simona Botti, Rita Bissani, Paola Casati, Alessandro Passera, Niccolò Miotti, Kris De Jonghe, Ellen Everaert, Antonio Olmos, Ana B. Ruiz-García, Francesco Faggioli

**Affiliations:** 1CREA Research Centre for Plant Protection and Certification, 00156 Rome, Italy; 2CAV-Centro Attività Vivaistiche, 48018 Faenza, Italy; 3Department of Agricultural and Environmental Sciences—Production, Landscape, Agroenergy, University of Milan, 20133 Milan, Italy; 4Flanders Research Institute for Agriculture, Fisheries and Food (ILVO), 9820 Merelbeke, Belgium; 5IVIA Instituto Valenciano de Investigaciones Agrarias, 46113 Valencia, Spain

**Keywords:** PLMVd, validated detection test, TPS, rapid-extraction methods

## Abstract

Peach latent mosaic viroid (PLMVd) is an important pathogen that causes disease in peaches. Control of this viroid remains problematic because most PLMVd variants are symptomless, and although there are many detection tests in use, the reliability of PCR-based methods is compromised by the complex, branched secondary RNA structure of the viroid and its genetic diversity. In this study, a duplex RT-qPCR method was developed and validated against two previously published single RT-qPCRs, which were potentially able to detect all known PLMVd variants when used in tandem. In addition, in order to simplify the sample preparation, rapid-extraction protocols based on the use of crude sap or tissue printing were compared with commercially available RNA purification kits. The performance of the new procedure was evaluated in a test performance study involving five participant laboratories. The new method, in combination with rapid-sample-preparation approaches, was demonstrated to be feasible and reliable, with the advantage of detecting all different PLMVd isolates/variants assayed in a single reaction, reducing costs for routine diagnosis.

## 1. Introduction

Peach latent mosaic viroid (genus *Pelamoviroid*, family *Avsunviroidae*) is an important pathogen that can seriously affect peaches (*Prunus persica*), inducing chlorotic mosaic or extreme albinism (calico disease) in leaves whilst the fruits turn misshapen and discolored with cracked sutures. Affected fruits become unmarketable, with yield reduction to a considerable extent. However, most PLMVd variants do not produce leaf symptoms, remain latent and can evolve to symptomatic variants and vice versa. It is present in all continents, and it is considered a quarantine pest in many countries (EPPO global database https://gd.eppo.int/search?k=peach+latent+mosaic+viroid (accessed on 26 April 2023) and IPPC website https://www.ippc.int/en/countries/all/regulatedpests/ (accessed on 26 April 2023)).

In Europe, in 2019, with Implementing Regulation (EU) 2019/2072, the European Commission established uniform conditions for protective measures against pests of plants. Peach latent mosaic viroid (PLMVd) was included in the list of pests for which visual inspection, and, in particular cases, sampling and testing are required for *Prunus persica* (Annex IV, part J).

PLMVd infection can induce a broad variety of symptoms that are often unstable and disappear seasonally. Visual inspection is therefore not always feasible to assess PLMVd infection. Moreover, most natural infections occur without apparent leaf symptoms, and often, symptoms need at least two years to develop [1]. The economic impact of PLMVd is mainly characterized by long-term effects of infection such as malformation and discoloration of fruits, which makes them unmarketable; reduced tree longevity; and increased susceptibility to other biotic and abiotic stresses [1]. PLMVd spreads mostly through infected propagation material and through infected pollen [2] that may be composed of mixed origins and therefore contain different isolates.

Rapid, effective and low-cost diagnostic tests are highly necessary to allow early and reliable PLMVd detection. Although molecular detection of PLMVd is effective, the rapidly evolving PLMVd genome accumulates changes that can potentially compromise a sequence-based detection assay [1,3,4].

Nonetheless, during the last decades, different methods for PLMVd detection were developed, using molecular hybridization [5], qualitative RT-PCRs [6], quantitative RT-PCRs [4,7,8] and loop-mediated amplification (LAMP) [9]. Even though all of those tests are reliable and robust, the quantitative RT-PCR method stood out as a diagnostic method over the past years because of its very high sensitivity, its speed and the redundancy of post-PCR manipulations, thereby minimizing chances for cross-contamination in the laboratory [10].

In this paper, a duplex reverse transcriptase–real-time PCR (dRT-qPCR) designed via merging two already published protocols [4,8] was developed and validated to overcome the problem of the high mutation rate of the PLMVd genome. The in silico analysis of the combination of primers and probes of the two tests already developed showed a very high inclusivity toward all of the isolates present in the database, as the complementarity of the two probes allows covering of the mismatches present in the genomic areas of the individuals. The duplex assay was tested in vitro on five different isolates of PLMVd and starting from different plant matrices. Moreover, the proposed assay was tested via comparing either different “classic” total RNA (TRNA) extraction protocol or “rapid” RNA extraction methods to simplify, speed up and decrease the cost of screening of peach trees. Finally, the performance of the duplex RT-PCR and the classic and rapid-extraction methods were evaluated in a test performance study (TPS) including five European laboratories.

## 2. Results

### 2.1. Development and Validation of the dRT-qPCR

Primers and probes already developed for PLMVd detection [4,8] were analyzed in silico to verify whether their merging in a single test was feasible. According to the multiple alignment obtained using both isolates reported in Table 1 and retrieved from GenBank (National Centre for Biotechnology information), the combination reported in Table 2 was selected.

The dRT-qPCR was first run on two PLMVd-infected samples (extracted according to [11]), and one isolate tested positive with both sRT-qPCRs, while the other was only detectable with the test by Serra et al. [4]; moreover, the duplex test was also run on a healthy sample. In all cases, the proposed duplex test gave the expected results (Figure 1).

Varying annealing temperatures and concentrations of probes were then applied to optimize such parameters in the dRT-qPCR test (data not shown). The best results were obtained at 58.5 °C of annealing temperature and at the concentration of probes reported in the Materials and Methods section.

The PLMVd-infected sample was then serially diluted tenfold, and the efficiencies of the sRT-qPCRs and the dRT-qPCR (Figure 2) were compared. Both sRT-qPCRs showed similar efficiencies, of 105%, which decreased to 96% when the probes were used in the dRT-qPCR (slope equal to −3.47).

The two sRT-qPCRs and the dRT-qPCR were also compared based on their ability to amplify TRNA extracted with the classic methods from phloem tissue of six PLMVd-infected samples (Table 1—Extraction tests). To overcome the bias introduced by different concentrations of the target in the six samples, the Cq values were normalized using the results obtained with the combination of TL and Zymo as a benchmark efficiency of the extraction. The mean ΔCq values were then compared as reported in Figure 3, where the normalized ΔCq values obtained for all classic extraction protocols are reported. According to the statistical analysis, no significant difference (*p* < 0.05) was observed among the Cq values obtained from the three detection methods using the same TRNA.

The dRT-qPCR was validated according to EPPO standard PM 7/98 [12]. Analytical sensitivity was assessed using three positive samples diluted tenfold in the total RNA of a healthy peach. The Limit of Detection (LOD) was assessed at a dilution of 10^−5^, which represents the last dilution in which all three samples gave positive Cq values (30 ± 1) [12] (Figure 4).

Analytical specificity was assessed as inclusivity and exclusivity. All 12 PLMVd-infected samples tested positive using the dRT-qPCR, confirming that the protocol has 100% inclusivity; considering that these samples belonged to different peach varieties, the selectivity of the test was also confirmed by the above-reported assays. Neither healthy peach samples nor nontarget isolates gave a positive reaction when tested with the dRT-qPCR (Table 3).

The dRT-qPCR was also tested for repeatability and reproducibility according to EPPO standard PM 7/98 [12]. Three samples (Table 1) were diluted at medium (10^−2^) and low (10^−4^) concentrations according to the results obtained for analytical sensitivity and then tested three times by the same operator simultaneously (repeatability). A portion of the diluted samples were later tested on a following day by a different operator (reproducibility). The results are shown in Figure 5. According to EPPO standard PM7/98, the repeatability calculation was 100% and the standard deviation of the obtained Cq was equal to ±0.37 for the samples at medium concentration and equal to ±0.86 for the samples at low concentration. In addition, the calculation of reproducibility was 100%, and the standard deviation increased up to ±1.20 for the samples at medium concentration and ±1.54 for the samples at low concentration. These values highlighted that the assay was highly repeatable and reproducible, both in absolute and in relative terms.

### 2.2. Extraction Tests

#### 2.2.1. Classic Extraction Methods

TRNA from six PLMVd-infected and four heathy peach samples (Table 1) were extracted using three classic extraction methods and twelve rapid-extraction methods (as combinations of different grinding buffers, types of membrane and release solutions—see Section 4.3).

As expected, all of the infected samples gave positive results, and no signals were obtained in the case of the healthy plants with any of the extraction methods. To compare the efficiency of the different extraction methods, the ΔCq values were evaluated, using as a benchmark the combination of Tissue Lyser and the Quick-RNA Plant Kit. As reported in Figure 6, the results from the Quick-RNA Plant Kit were comparable to those obtained with the RNeasy Plant mini-kit using both liquid nitrogen and Tissue Lyser. The ΔCq values from the TRNA extracted with the Sbeadex maxi-plant kit were statistically different from those of the other kits (*p* < 0.001). In this case, the values obtained with the combination of Tissue Lyser and the Sbeadex maxi-plant kit turned out better (statistically significant, *p* < 0.001) than with the use of liquid nitrogen.

#### 2.2.2. Rapid-Extraction Methods

Tissue print and crude extracts spotted on paper filters or nylon membranes were tested as rapid-extraction methods [13,14]; different buffers were used for maceration (PBS and PO_4_ [15]) and for nucleic-acid recovery (glycine buffer and triton X-100) [13,14,16].

For the obtained results, the statistical analysis did not highlight a single parameter (grinding buffer, type of blotting tissue, release solution) determining for itself significant differences in extraction performance (data not shown). However, significant differences were obtained when the variables were all analyzed together and all of the different combinations were taken into account (Figure 7). Specifically, the tissue prints showed significantly the best results (*p* < 0.001), using nylon membranes and glycine as the releasing solution compared to other combinations (Figure 7a). Minor statistical diversity was found in analysis of crude sap spotted on membranes. Small differences were highlighted when grinding was made with the PBS buffer (Figure 7b); in this case, all of the results did not deviate, except for the filter paper/triton combination with respect to the nylon membrane/glycine (*p* < 0.05). The results using the PO_4_ buffer were all similar, but the nylon/glycine combination seemed to decrease the efficiency in amplification (*p* < 0.001; Figure 7c).

The best combinations were obtained with the phloem matrix, as the tissue printing/nylon membrane/glycine buffer and PBS maceration/nylon membrane/glycine buffer combinations were tested also on leaves (Figure 8). The results obtained on the leaf tissue were comparable with those obtained starting from phloem using nylon membranes and a triton buffer. Leaf tissues were also macerated with the DAS-ELISA extraction buffer, with obtained results comparable to those of the other rapid-extraction methods.

### 2.3. Test Performance Study (TPS)

Five European laboratories took part in the TPS:Centro attività vivaistiche (CAV), ItalyCREA—Centro di Ricerca Difesa e Certificazione (CREA-DC), Italy—Organizing LaboratoryDipartimento di Scienze Agrarie e Ambientali—Produzione, Territorio, Agroenergia, Università degli studi di Milano (UNIMI), ItalyFlanders Research Institute for Agriculture, Fisheries and Food (ILVO), BelgiumInstituto Valenciano de Investigaciones Agrarias (IVIA), Spain

Healthy samples, PLMVd-infected samples and one nontarget sample (a peach sample infected by a different target) (Table 1 and Table 3) were prepared for the TPS. The organizing laboratory included, in the panel, samples extracted with a conventional kit (Set A); leaf samples ground using the DAS-ELISA (Bioreba) extraction buffer (Set B); phloem tissues printed on nylon membranes (Set C); and phloem tissues ground in a PBS buffer and spotted on nylon membrane (Set D).

All sample sets were randomized and the participants anonymized before shipping. Ten percent of the samples for each sample item were analyzed for homogeneity before shipping and for stability after all of the participating laboratories submitted their results (Table 4).

All participants were able to submit the results on time. Due to the loss of a tube containing resuspension solution, the participant Lab 4 was not able to perform the analysis on Set B. The participant Lab 3 made a deviation from the proposed protocol, using a different master mix for the dRT-qPCR amplification (One Step PrimeScript RT-PCR kit supplied by Takara).

A positive amplification control (PAC) was provided along with the sample panel. All of the participants submitted results for the PAC. The PAC results were used for the first quality check of the analyzed data sets. An average Cq value of 20.5 ± 2.8 was obtained, confirming the good quality of the data sets.

Table 5 reports the results obtained in analysis of Set A as they were submitted by each participant. Some non concordant data were highlighted due to false positives or undetermined results; through comparing the Cq values obtained by all participants, it is clear that some laboratories applied a Cq cut-off value, while others instead reported all of the samples that produced an exponential curve over the baseline as positive or undetermined.

Since a clear separation of the Cq values obtained for the positive and negative samples could be easily observed by checking the Cq values obtained by the participants, the application of a correct cut-off helped in making a decision regarding the “undetermined” results.

For the samples spotted on membranes, the results are reported in Table 6.

Compared to the TRNA extracted using a conventional kit, there were no false-positive results, only some false-negative results. The false negatives were mostly from only one participant (Lab 3); this could be due to the deviation in protocol made by Lab 3, which used a different master mix for the dRT-qPCR amplification (the One Step PrimeScript RT-PCR kit supplied by Takara instead of the TaqMan™ RNA-to-CT™ 1-Step Kit by ThermoFisher Scientific, Waltham, MA, USA). Most likely, the samples subjected to rapid extraction, especially those involving nylon membranes, were more difficult to amplify; hence, even small changes in protocol could affect the results. For this reason, results from Lab 3 (Sets C and D) were not taken into account in the calculation of the performance of the protocol [17].

According to the above considerations, the performance criteria (Table 7) were calculated using the formulas reported in EPPO standard PM 7/122 [18] and by Massart et al. [17].

The performance criteria for Set A were calculated considering both results with (*) and without the application of a cut-off.

## 3. Discussion

Control and diagnosis of PLMVd is dependent on several factors because detection by visual inspection is limited to the appearance of symptomatology and PCR detection methods are hampered by the high variability of the viroid genome and the complex, branched secondary RNA structure. Although reliable diagnostic methods have been developed, some PLMVd variants often go undetected due to the genetic variability of the viroid. Analysis of the PLMVd sequences available in public databases and analysis of previously published PCR-based methods revealed primers and probes able to detect different representative variants [4]. In order to develop a single method with the potential to detect all isolates, combinations of candidate primers and probes were used. This paper reports the development of a duplex RT-qPCR system (dRT-qPCR), its comparison with the single real-time amplifications (sRT-qPCRs) and an evaluation of several classic or rapid nucleic-acid-extraction methods in order to optimize PLMVd detection. All of the classic RNA extraction methods resulted suitable for PLMVd detection, although the best results were obtained when silica-column-based kits were used; further optimization will be needed to efficiently use magnetic-bead-based kits due to the peculiar physicochemical characteristics of viroids. Several rapid nucleic-acid-extraction protocols were tested; all of them were able to discriminate healthy and PLMVd-infected plants. Rapid extractions of phloem tissue were able to detect PLMVd over the whole season. Rapid extractions of leaf tissue were also feasible for accurate diagnosis, with the DAS-ELISA extraction buffer used as a grinding buffer. The possibility of using these reliable rapid-extraction methods makes PLMVd analysis faster and cheaper.

The new diagnostic method of dRT-qPCR has been validated according to EPPO standard PM 7/98 [12]. The validation data obtained (analytical sensitivity, analytical specificity, repeatability and reproducibility) were comparable with those of the individual tests, confirming the reliability of the duplex test. As expected, analytical specificity was higher for the developed method that was able to identify variants not detected with the sRT-qPCR, improving the diagnostic methods previously published due to the ability to detect all of the variants in only one reaction. Moreover, the new dRT-qPCR assay (including the extraction methods) was evaluated with a TPS among five laboratories. The TPS results were very encouraging; in fact, the duplex RT-qPCR demonstrated high reproducibility with all of the different extraction methods tested. However, some considerations certainly emerged from a careful analysis of the results. The TPS showed that in some cases, there were problems with high Cq values for some healthy samples due to nonspecific amplification. These problems could be overcome if the correct cut-off value were applied by the participants: an evaluation that could be made only after the protocol was used routinely by the laboratory. It is interesting to note that the laboratories reporting these problems (Labs 1, 2 and 5) did not report issues with shipping; instead, Lab 3 especially highlighted that samples arrived particularly warm. This could also help with better understanding of the reason for noncompliant results.

Here, we provide evidence that rapid-extraction methods can provide reliable-enough results to be used in routine high-throughput diagnostics, preventing the spread of PLMVd in propagation materials and in the field. Specifically, the protocol used for Panel Set B could be especially feasible when the certification process is ongoing because it uses the same sample preparation made for the DAS-ELISA for fruit-tree viruses. The protocol of Set C could be easily performed directly in the field, avoiding collecting of large volumes of samples. The protocol of Set D seems the most reliable; it could be very useful in performing PLMVd detection in plant tissues with low viroid concentrations, hence, through all of the vegetative seasons, as well as in the absence of leaves during winter dormancy.

In conclusion, the new and validated duplex RT-PCR assay opens new possibilities in the prevention, control and epidemiology studies of PLMVd, with potential to be used for identification of unknown vectors. One of the advantages of this method based on RT-PCRs is the possibility to use it in combination with simple sample-preparation methods due to its sensitivity. This result has an interesting practical implication, since in a broad-spectrum diagnostic analysis of peach trees (for example, in the context of the propagation-material certification process), it is possible to test the main peach viruses with the ELISA and PLMVd with the RT-qPCR while starting from the same ground material.

## 4. Materials and Methods

### 4.1. Sample Collection and TRNA Extraction

Fourteen samples with defined phytosanitary statuses were obtained from the collection of CREA-DC in Rome. Moreover, 10 other peach samples were collected from commercial orchards in the Lazio region (Central Italy): 5 infected with PLMVd and 5 infected with other virus and viroid species (Table 1). TRNA was extracted according to the procedure reported by Luigi and Faggioli [11], starting from phloem and leaf tissue and using Tissue Lyser with McKenzie buffer [20] for homogenization and the Quick-RNA Plant Kit (Zymo Research, CA, USA) or the RNeasy Plant mini-kit (Qiagen, Germany). Some samples were collected in autumn and spring, as reported in Table 1.

### 4.2. Duplex RT-qPCR: Development and Validation

The primers and probes published by Luigi and Faggioli and Serra et al. [4,8] were evaluated in silico for their characteristics (annealing temperature, self-dimer and cross-dimer check) and aligned on the PLMVd genomes (Table 2), and the best combination was used in the duplex test. The primer RP2 was slightly modified, with three nucleotides added to the 3′ end to increase its melting temperature (reported in bold in Table 2). Both probes were included in the test with the same fluorophore to overcome problems of efficiency.

The dRT-qPCR was optimized for the relative concentrations of the probes and the temperatures of the annealing/extension steps. The optimal reaction conditions were as follows: 1 μL of target RNA was added to 9 μL of the reaction mixture based on the use of a TaqMan™ RNA-to-CT™ 1-Step Kit (ThermoFisher Scientific, Waltham, MA, USA). Briefly, the reaction mixture was as follows: 1× master mix, 1× RT enzyme mix, 0.5 μM of each primer, 0.4 μM of the P3 probe and 0.5 μM of the PLMVd-P probe.

cDNA was synthesized for 15 min at 48 °C, followed by 10 min of denaturation at 95 °C. Amplification was performed as follows: denaturation at 95 °C for 15 s, annealing and extension at 58.5 °C for 1 min, for a total of 40 cycles. The assays were carried out on a CFX96 Touch system (BioRad, Hercules, CA, USA).

The efficiencies of the optimized dRT-qPCR and of the two sRT-qPCRs were compared using tenfold serial dilutions of samples already used for test development (phloem tissue, collected in autumn—Table 1); sRT-PCRs were performed using the RNA-to-CT™ 1-Step Kit (ThermoFisher Scientific) with primers and probe concentrations according to the respective publications. Standard curves were obtained by plotting the Cq values of the tenfold dilution series versus the logarithm of the dilution factor. The following equation [21] was used to determine the efficiency (*E*) of each amplification from the slope of the linear regression model; the linear correlation coefficient (R^2^) was also reported:$$E\left(\%\right)=\left(10^{1-slope}-1\right)\times 100$$

Then, the dRT-qPCR was validated according to EPPO standard PM 7/98 [12].

Analytical sensitivity was assessed by measuring the Cq values of 6 tenfold serial dilutions of three PLMVd isolates in TRNA from healthy peaches. Analytical specificity was considered as inclusivity and exclusivity. Inclusivity was assessed via testing 12 different PLMVd-infected peach trees (*P. persica*) belonging to different cultivars (Table 1). Exclusivity was evaluated via testing the most important viruses/viroids that affect peaches according to European legislation (Implementing Regulation (EU) 2019/2072), i.e., apple chlorotic leaf spot virus (ACLSV), apple mosaic virus (ApMV), apple stem grooving virus (ASGV), apple stem pitting virus (ASPV), prunus dwarf virus (PDV), prunus necrotic ringspot virus (PNRSV), plum pox virus (PPV) strains D and M and hop stunt viroid (HSVd). Repeatability and reproducibility were assessed in-house through testing three samples at medium and low concentrations. Repeatability was assessed via performing the test simultaneously. Reproducibility was assessed using a portion of the same samples tested for repeatability but at different times and with different operators.

### 4.3. Extraction Tests

Different extraction methods, both classic and rapid, were compared. For the classic methods, two procedures were applied:An amount of 0.1 g of fresh phloem tissue was added to 1 mL of lysis buffer [20] containing 2% of sodium metabisulfite and disrupted using Tissue Lyser (TL, Qiagen) at maximum speed (30 Hz) for 5 min (using three beads for a sample).An amount of 0.1 g of fresh phloem tissue was homogenized with a mortar and pestle in liquid nitrogen (N_2_) and lysed using 1 mL of lysis buffer [20] already added with 2% sodium metabisulfite.The tubes were then centrifuged and 1 mL of supernatant collected, added with 100 μL of 20% N-Lauroylsarcosine sodium salt solution and incubated for 5 min at 70 °C; then, the TRNA was extracted using:(a)The Quick-RNA Plant Kit, according to the manufacturer’s instructions;(b)The RNeasy Plant mini-kit, according to the manufacturer’s instructions;(c)The Sbeadex maxi-plant kit (Biosearch technologies, Hoddesdon, UK) in combination with the King Fisher (ThermoFisher) automation system, according to the manufacturers’ instructions.

Finally, the TRNA was amplified in two technical replicates, both with the dRT-qPCR and the two single RT-qPCRs (in the conditions reported above).

Regarding the rapid-extraction method, some procedures, already applied for other viruses and viroids, were combined according to the scheme reported in Figure 9.

Specifically, phloem and leaf tissue samples were directly printed on membranes (both paper—Whatman 3 MM and nylon membrane—Roche [13,14]) or spotted as crude extracts using two buffers, PBS buffer supplemented with 2% polyvinylpyrrolidone (PVP) and 0.2% sodium diethyl dithiocarbamate (DETC) [15] and PO_4_ buffer (Na_2_HPO_4_/KH_2_PO_4_ 0.1 M pH 7.2), both used at a 1:10 *w*/*v*. The crude extracts were then centrifuged for 3 min at 6000 rpm, and 5 µL of supernatant was spotted on 5 mm-diameter filter papers or nylon membranes previously inserted in 1.5 mL tubes and left to dry. Nucleic acid from each membrane (tissue-printed or spot-blotted) was retrieved using 100 µL of 0.5% triton X-100 [13] or glycine buffer (0.1 M glycine, 0.05 M NaCl, 1 mM EDTA) [14,16]. All samples were amplified with the dRT-qPCR assay in two technical replicates.

Leaf samples were also spotted as crude extracts using the ELISA extraction buffer (Bioreba, Switzerland) for grinding.

### 4.4. Statistical Analysis

Statistical analyses were performed using R software, version 4.1.1 [19]. Raw data, consisting of Cq values of templates obtained from the different extractions, were normalized by the respective Cq values obtained by Tissue Lyser and Quick-RNA Plant Kit extraction, which was considered a benchmark protocol. Normalized data were presented as ΔCq values.

ΔCq values of dRT-qPCRs obtained when testing classic and rapid-extraction methods were statistically compared and analyzed with one-way ANOVA followed by Tukey’s “Honest Significant Difference” method.

Those that did have abnormal distributions were hence compared with the Kruskal–Wallis test followed by Tukey’s Honest Significant Difference (HSD) post hoc test.

### 4.5. Test Performance Study

In order to organize the test performance study (TPS), samples (leaf and phloem tissues) were collected during spring of 2022 and split into ten panels. Each panel was composed of four sets of six blind samples and one positive amplification control, each set including two healthy, one nontarget and three PLMVd-infected samples randomized for each participant:

Set A—Phloem tissue, extracted with Tissue Lyser and the Quick-RNA Plant Kit (Zymo Research) and spotted on filter paper (Whatman), to be resuspended in 100 µL of DePC water;

Set B—Leaf tissue, ground in ELISA extraction buffer (Bioreba) and spotted on filter paper (Whatman), to be resuspended in 100 µL of Triton X solution;

Set C—Phloem tissue, directly printed on nylon membrane, to be resuspended in 100 µL of glycine buffer;

Set D—Phloem tissue, macerated in PBS buffer and spotted on nylon membrane, to be resuspended in 100 µL of glycine solution.

Five European laboratories participated in the TPS: Centro Attività Vivaistiche (CAV), Faenza, Italy; Flanders Research Institute for Agriculture, Fisheries and Food (ILVO), Merelbeke, Belgium; Instituto Valenciano de Investigaciones Agrarias (IVIA), Valencia, Spain; Department of Agricultural and Environmental Sciences—Production, Landscape, Agroenergy—University of Milan, Italy; and CREA—Research Centre of Plant Protection and Certification (CREA-DC), Rome, Italy. Ready-to-use mixtures of primers and probes and the four solutions for nucleic-acid resuspension were also provided to the participants. Performance criteria and validation procedures were established following guidelines from EPPO standards PM 7/98 [12] and PM 7/122 [18]; repeatability and reproducibility were calculated applying the method from Langton et al. [22].

## Figures and Tables

**Figure 1 plants-12-01802-f001:**
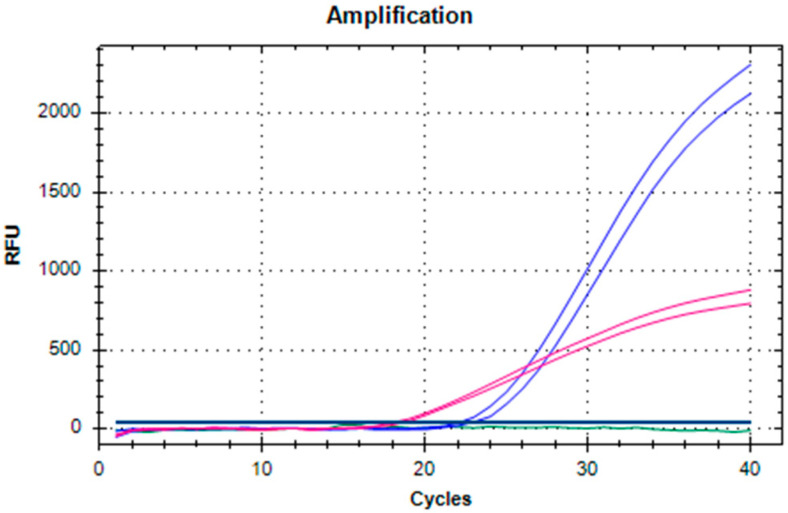
Example of an amplification plot of dRT-qPCR for PLMVd. Blue curves: PLMVd-positive sample (ID 2 of Table 1) that tested positive with both sRT-PCRs; pink curves: PLMVd sample (ID 1 of Table 1) only detectable with the test by Serra et al. [4]; green curve: healthy sample.

**Figure 2 plants-12-01802-f002:**
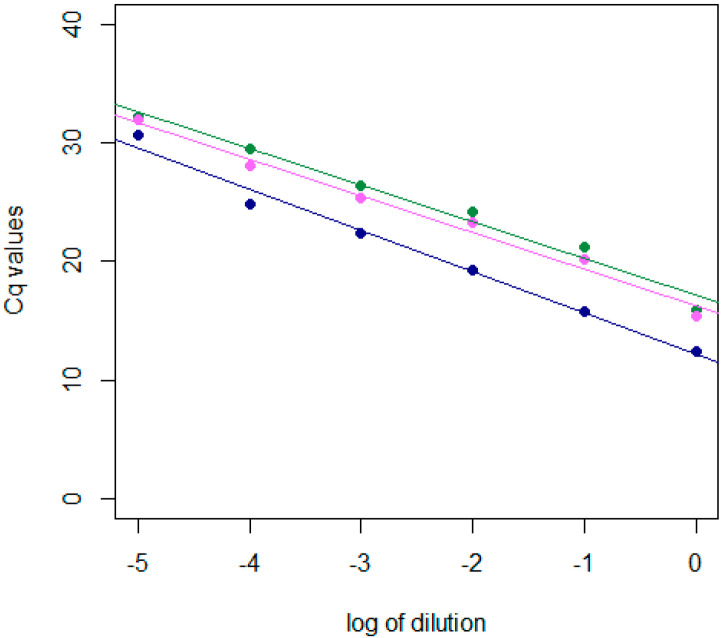
This graph highlights the efficiency-curve comparison. The horizontal axis reports the logarithm of the dilution factor, while the vertical axis shows the Cq values obtained in the reactions. In blue are the points and the interpolating line of the dRT-qPCR (slope = −3.41; R^2^ = 98%); in green are the points and the interpolating line of the sRT-qPCR by Luigi and Faggioli [8] (slope = −3.10; R^2^ = 99%); and in purple are the points and the interpolating line of the sRT-qPCR by Serra et al. [4] (slope = −3.10; R^2^ = 98%).

**Figure 3 plants-12-01802-f003:**
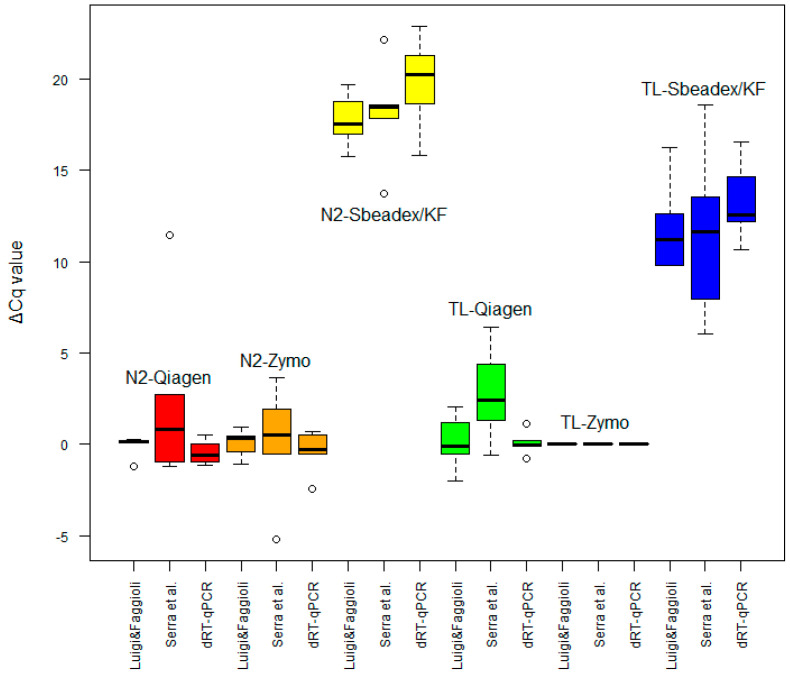
ΔCq values of amplifications using three different real-time PCR tests: Luigi and Faggioli [8], Serra et al. [4] and the dRT-qPCR developed in this work. Each test was applied to six different classic extraction protocols: liquid nitrogen (N2) + RNeasy Plant mini-kit (Qiagen)—red; liquid nitrogen (N2) + Quick-RNA Plant Kit (Zymo)—orange; liquid nitrogen (N2) + Sbeadex maxi-plant kit (Sbeadex/KF)—yellow; Tissue Lyser (TL) + RNeasy Plant mini-kit (Qiagen)—green; Tissue Lyser (TL) + Quick-RNA Plant Kit (Zymo)—purple; and Tissue Lyser (TL) + Sbeadex maxi-plant kit (Sbeadex/TL)—blue. Cq values were normalized (ΔCq) using the Tissue Lyser (TL) + Quick-RNA Plant Kit (Zymo) as a benchmark. Values are expressed as boxplots of six PLMVd-infected plants (two technical replicates).

**Figure 4 plants-12-01802-f004:**
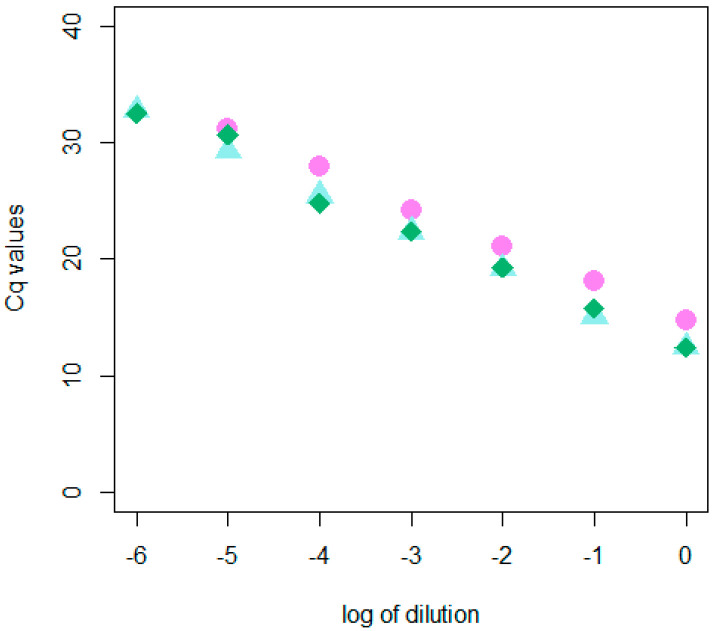
Cq values obtained by testing three PLMVd isolates at relative tenfold dilution (reported as logarithm).

**Figure 5 plants-12-01802-f005:**
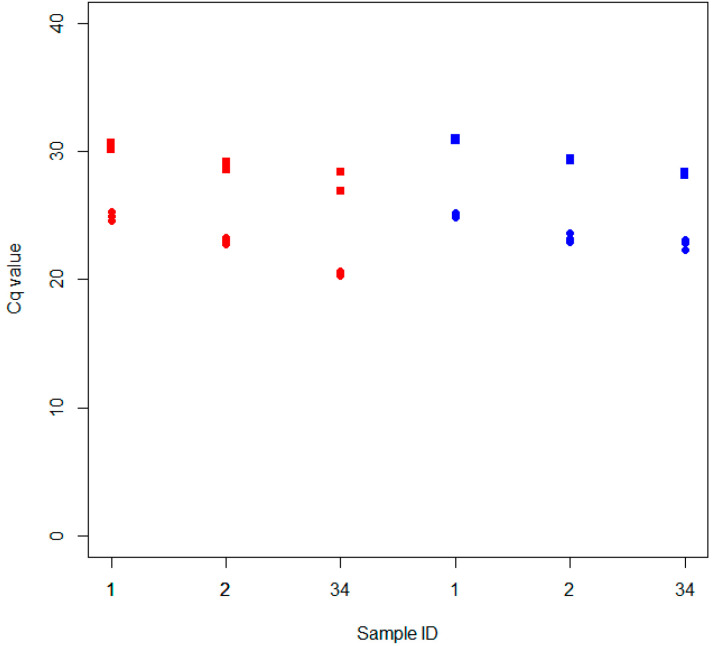
Cq values obtained by testing the repeatability (red) and reproducibility (blue) of the dRT-qPCR. Samples were tested at low (■) and medium (●) concentrations, each in three technical replicates.

**Figure 6 plants-12-01802-f006:**
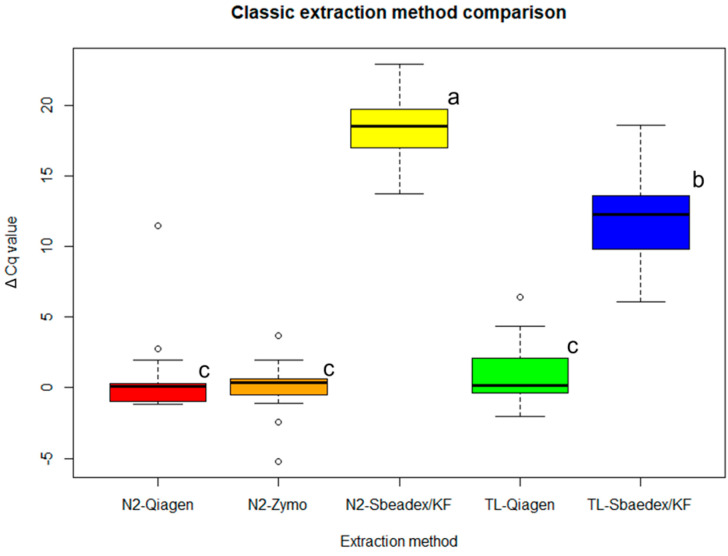
Comparison of the mean ΔCq values obtained in analyzing TRNA extracted with the following combinations: liquid nitrogen + RNeasy Plant mini-kit—red; liquid nitrogen + Quick-RNA Plant Kit—orange; liquid nitrogen + Sbeadex maxi-plant kit—yellow; Tissue Lyser + RNeasy Plant mini-kit—green; Tissue Lyser + Sbeadex maxi-plant kit—blue. Statistical significance of differences was determined using Tukey’s HSD post hoc test; different letters indicate statistically different groups (*p* < 0.001).

**Figure 7 plants-12-01802-f007:**
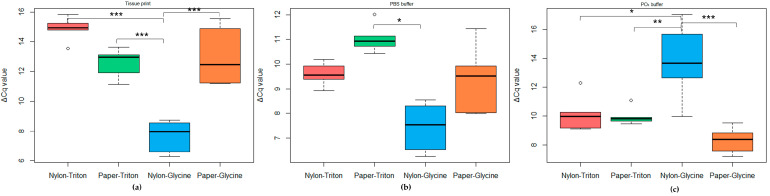
Boxplot reporting the ΔCq values obtained in analysis of TRNA from rapid extraction: (**a**) results obtained in analysis of tissue-printed samples; (**b**) results obtained in analysis of samples macerated in PBS buffer; and (**c**) results obtained in analysis of samples macerated in PO_4_ buffer. Statistical significance of different ΔCq values was determined using Tukey’s HSD post hoc test (* = *p* < 0.05; ** = *p* < 0.01; *** = *p* < 0.001).

**Figure 8 plants-12-01802-f008:**
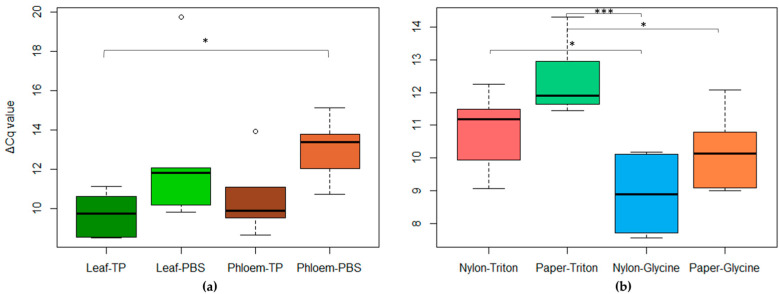
(**a**) Boxplot reporting the ΔCq values obtained in analysis of leaf samples vs phloem samples through tissue printing (TP) or maceration in PBS (PBS). (**b**) Comparison of the ΔCq values of the leaf samples macerated in Bioreba buffer, spotted on nylon or paper membranes and released with Triton or glycine buffer. Statistical significance of differences was determined using Tukey’s HSD post hoc test (* = *p* < 0.05; *** = *p* < 0.001).

**Figure 9 plants-12-01802-f009:**
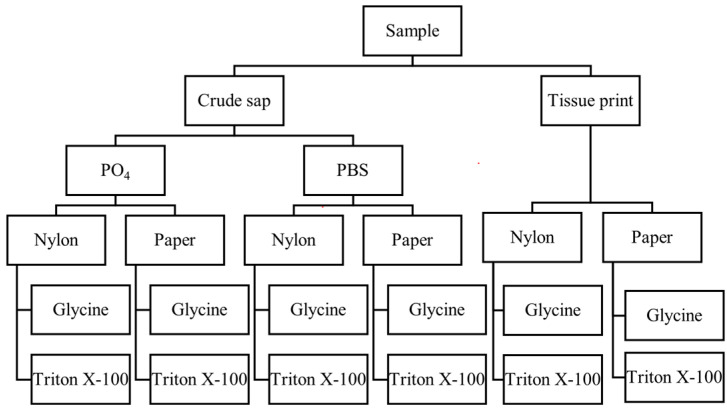
Graphical representation of the different combinations of rapid-extraction tests used.

**Table 1 plants-12-01802-t001:** List of the samples collected to develop and validate the test, with their phytosanitary statuses. For each sample, type of matrix (P for phloem and L for leaf), date of sampling (A for autumn and S for spring), host and steps of the workflow in which it was used are reported. With a *, isolates were retrieved from a commercial orchard.

Sample ID	Phytosanitary State	Matrix	Period of Sampling	GenBankAcc. No.	Host	Test Development	Analytical Sensitivity	Analytical Specificity	Repeatability/Reproducibility	Extraction Tests	TPS
*1*	PLMVd	P	A; S	ON513442	*P. persica*		X	X	X		
*2*	PLMVd	P, L	A; S	ON513443	*P. persica* cv Opera	X	X	X	X	X	X
*22*	PLMVd	P, L	A; S	ON513444	*P. persica* cv Rosa del West		X	X		X	X
*37*	PLMVd	P, L	A; S	ON513445	*P. persica* cv Zaigadi Royal Jim^®^			X		X	
*43*	PLMVd	P, L	A; S	ON513446	*P. persica* cv Zaisito Patty^®^			X		X	X
*42*	PLMVd	P, L	A; S		*P. persica* cv Nerid01206 Romagna sweet^®^			X		X	
*34*	PLMVd	P, L	A; S		*P. persica* cv Alma			X	X	X	
*M54 **	PLMVd	P	S		*P. persica* cv Tardiva di San Vittorino			X			
*M56 **	PLMVd	P	S		*P. persica* cv Crasiomolo			X			
*M57 **	PLMVd	P	S		*P. persica* cv Crasiomolo			X			
*M58 **	PLMVd	P	S		*P. persica* cv Crasiomolo			X			
*M59 **	PLMVd	P	S		*P. persica* cv Reginella II			X			
*NT1 **	ACLSV	P	S		*P. persica*			X			
*NT2 **	ApMV	P	S		*P. persica*			X			
*NT3 **	ASGV/ASPV	P	S		*Pomaceae*			X			
*NT4 **	PDV	P	S		*Prunus* spp.			X			
*NT5 **	PNRSV	P, L	S		*P. persica*			X			X
*6.3*	PPV strain D	P	S		GF305			X			
*7.3*	PPV strain M	P	S		GF305			X			
*CMC D*	HSVd	P	S		*Citrus* spp.			X			
*PPE42*	Healthy	P	A; S		*P. persica*	X		X		X	
*PPE44*	Healthy	P	A; S		*P. persica*			X		X	
*PPE60*	Healthy	P	A; S		*P. persica*			X		X	X
*PPE80*	Healthy	P	A; S		*P. persica*			X		X	X

**Table 2 plants-12-01802-t002:** List of the primers and probes used for the setup of the dRT-qPCR. In bold, the nucleotides added to the published sequence.

Name	Sequence (5′-3′)	Position	Reference	Used in the Duplex
**PLMVd-P**	FAM-CTTCTGGAACCAAGCGG-BHQ1	165–181	[8]	Yes
**PLMVd-H**	CTCGCAATGAGGTAAGGTG	137–155		No
**PLMVd-C**	ACGTCGTAATCCAGTTTCTAC	236–216		No
**P3**	FAM-GGTACCGCCGTAGAAACTGGGTTACG-BHQ1	207–232	[4]	Yes
**RP2**	GGGACCGGGWTTGAAT**CCG**	261–246		Yes (modified)
**FP2**	CAATGASGTAAGGTGGGACT	141–160		Yes

**Table 3 plants-12-01802-t003:** List of the samples collected to ascertain inclusivity and exclusivity of the test, with their phytosanitary states, the hosts and the results obtained.

	Phytosanitary State	Host	Result
Inclusivity	PLMVd	*P. persica* (GF365)	Positive
	PLMVd	*P. persica* cv Opera	Positive
	PLMVd	*P. persica* cv Rosa del West	Positive
	PLMVd	*P. persica* cv Alma	Positive
	PLMVd	*P. persica* cv Zaigadi Royal Jim^®^	Positive
	PLMVd	*P. persica* cv Nerid01206 Romagna Sweet^®^	Positive
	PLMVd	*P. persica* cv Zaisito Patty^®^	Positive
	PLMVd	*P. persica* cv Tardiva di San Vittorino	Positive
	PLMVd	*P. persica* cv Crasiomolo cl. B (Gialla spicca)	Positive
	PLMVd	*P. persica* cv Crasiomolo cl. C (Duracina)	Positive
	PLMVd	*P. persica* cv Crasiomolo cl. C (Duracina)	Positive
	PLMVd	*P. persica* cv Reginella II	Positive
Exclusivity	ACLSV	*P. persica*	Negative
	ApMV	*P. persica*	Negative
	ASGV/ASPV	*Pomaceae*	Negative
	PDV	*Prunus* spp.	Negative
	PNRSV	*P. persica*	Negative
	PPV strain D	GF305	Negative
	PPV strain M	GF305	Negative
	HSVd	*Citrus* spp.	Negative
	Healthy	*P. persica*	Negative
	Healthy	*P. persica*	Negative
	Healthy	*P. persica*	Negative
	Healthy	*P. persica*	Negative

**Table 4 plants-12-01802-t004:** Mean Cq values and standard deviations calculated by analyzing three technical repetitions of each sample item before (homogeneity) and after (stability) the shipment.

Sample ID	Homogeneity				Stability			
	Set A	Set B	Set C	Set D	Set A	Set B	Set C	Set D
**PPE60**	>35	>39	>38	>39	>35	>39	>38	>39
**PPE80**	>35	>38	>38	NA	>35	>39	>38	>37
**NT5**	>35	>36	>38	>37.5	>36	>38	>37	>37
**2**	14.9 ± 0.9	28.9 ± 1.7	23.7 ± 1.9	24.9 ± 0.8	17.6 ± 0.6	32.4 ± 0.8	24.6 ± 0.6	25.3 ± 0.5
**22**	16.7 ± 1.2	29.9 ± 0.5	27.0 ± 2.4	29.7 ± 0.7	20.4 ± 1.9	31.8 ± 0.6	24.8 ± 0.7	28.7 ± 0.3
**37**	16.5 ± 1.1	28.3 ± 1.3	30.0 ± 0.7	30.9 ± 0.4	18.9 ± 1.0	30.9 ± 3.0	24.5 ± 0.1	27.8 ± 1.0

**Table 5 plants-12-01802-t005:** Comparison of the results submitted by each participant and the relative mean Cq values obtained in analysis of sample items from Set A. In red are the non concordant results.

		LAB1		LAB2		LAB3		LAB4		LAB5	
**Set A**	PPE60	Und	34.34	Und	33.62	Pos	39.16	Neg	39.12	Neg	35.39
	PPE80	Pos	33.51	Und	34.17	Pos	38.46	Neg	39.06	Neg	35.29
	NT5	Pos	32.45	Neg	34.25	Neg	40.00	Neg	40.00	Neg	34.81
	2	Pos	15.76	Pos	17.37	Pos	22.20	Pos	22.55	Pos	14.20
	22	Pos	24.81	Pos	18.41	Pos	20.22	Pos	27.30	Pos	15.87
	37	Pos	18.48	Pos	18.00	Pos	21.96	Pos	23.89	Pos	15.73

**Table 6 plants-12-01802-t006:** Comparison of the results submitted by each participant and the relative mean Cq values obtained in analysis of Sets B, C and D. In red are the non concordant results.

	LAB1			LAB2			LAB3			LAB4			LAB5		
Sample/Set	B	C	D	B	C	D	B	C	D	B	C	D	B	C	D
**PPE60**	Neg	Und	Neg	Neg	Neg	Neg	Neg	Neg	Neg	NT	Neg	Neg	Neg	Neg	Neg
**PPE80**	Neg	Neg	Neg	Neg	Neg	Neg	Neg	Neg	Neg	NT	Neg	Neg	Neg	Neg	Neg
**NT5**	Neg	Neg	Neg	Und	Neg	Neg	Neg	Neg	Neg	NT	Neg	Neg	Neg	Neg	Neg
**2**	Pos	Pos	Pos	Pos	Pos	Pos	Pos	Neg	Neg	NT	Pos	Pos	Pos	Pos	Pos
**22**	Pos	Pos	Pos	Pos	Pos	Pos	Pos	Neg	Neg	NT	Neg	Pos	Pos	Pos	Pos
**37**	Pos	Pos	Pos	Pos	Pos	Pos	Neg	Neg	Neg	NT	Pos	Pos	Pos	Pos	Pos

**Table 7 plants-12-01802-t007:** Performance criteria obtained in the TPS. * Indicates the results of applying a cut-off for Set A.

		SET A	SET A*	SET B	SET C	SET D
**Total data set**		5	5	4	4	4
**Total data points**	N	30	30	24	24	24
**True positive**	TP	15	15	11	11	12
**True negative**	TN	8	15	11	11	12
**False positive**	FP	4	0	0	0	0
**False negative**	FN	0	0	1	1	0
**Concordant**	TP+TN	23	30	23	23	24
**Non concordant**	FP+FN	4	0	1	1	0
**Accuracy (%)**	(TP+TN)/N	77%	100%	96%	96%	100%
**Diagnostic sensitivity (%)**	TP/(TP+FN)	100%	100%	92%	92%	100%
**Diagnostic specificity (%)**	TN/(TN+FP)	67%	100%	100%	100%	100%
**Reproducibility (%)**	Langton et al. [19]	75%	100%	88%	91%	100%

## Data Availability

Data will be available on request.

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
