# Peer review of "Development and Validation of a Duplex RT-qPCR for Detection of Peach Latent Mosaic Viroid and Comparison of Different Nucleic-Acid-Extraction Protocols"

_plants, 2023, doi:10.3390/plants12091802_

Round 1

Reviewer 1 Report

Minor comments and suggestions indicated per line. Mostly quoted text and suggested replacement. 

L42: ".... uniform conditions as regards protective measures..." 

".... uniform conditions for protective measures..."

L46-48: "Visual inspection is not always feasible to assess an infection with PLMVd because even if PLMVd can induce a broad variety of symptoms, those are often not stable and disappear with time. "

"PLMVd infection can induce a broad variety of symptoms, that are often unstable and disappear during the season. Visual inspection is therefore not always feasible to assess PLMVd infection."

L49-51: "The economic impact of PLMVd is mainly characterized by long term problems, like fruits alteration, reduced tree longevity, and increased susceptibility to other biotic and abiotic stresses [1]."

 "The economic impact of PLMVd is mainly characterized by long term effects of infection such as, the alteration of fruit, reduced tree longevity, and increased susceptibility to other biotic and abiotic stresses [1]."

Also: if what way is the fruit altered? Maybe expand or clarify the statement.

L54-57: "In that respect, rapid, effective and low-cost diagnostic tests are highly necessary to allow an early and reliable PLMVd detection. Generally, viroid detection can be tricky and due to the lack of the coat protein there is the consequent impossibility to make use of massive screening test like ELISA. While molecular detection of PLMVd is effective, it  is made more difficult by the intrinsic property to evolve rapidly of PLMVd, quickly accumulating changes in the genome [1,3,4]. " 

"Rapid, effective and low-cost diagnostic tests are highly necessary to allow an early and reliable PLMVd detection. Generally, viroid detection can be tricky and due to the lack of the coat protein there is the consequent impossibility to make use of massive screening test like ELISA. Although molecular detection of PLMVd is effective, the rapidly evolving PLMVd genome accumulates changes that can potentially compromise a sequence-based detection assay [1,3,4]. "

L275: "...variants often escaped to the detection ..."

"...variants often go undetected ..."

L289: "...all the year." 

"...year round." or "...over the whole season"

L295: "...higher by the developed... "

"...higher for the developed... "

L309-312: "Another important consideration is that the result obtained in using the rapid extraction methods is very interesting. In fact, they proved to be reliable enough to be used in massive analysis, thus facilitating a wider, earlier and economic control, preventing the spread of PLMVd in propagation material and in field. "

"Here we provide evidence that rapid extraction methods can provide reliable enough results to be used in routine high-throughput diagnostics, preventing the spread of PLMVd in propagation material and in field. "

QUESTIONS FOR CLARIFICATION

L97: Please clarify which RNA extract was used for this optimisation?

L111: Please provide more information on how the normalisation was performed.

L167: Why no Zymo TL comparison?

L176: consider starting a new paragraph with a separate subheading.

L193-195: Wording can be confusing. Consider re-phrasing.

Figure 7: Please add headings to graphs with extraction method.

Figure 8: Why are these two graphs in the same figure? 

a) Leaf vs phloem comparison. Not clear that hose were the two best methods depicted in Fig7. 

b) This is a nylon vs paper comparison. But now with Bioreba buffer?  

Somewhere I am missing the link.

L216: Sample? Does this refer to the RNA extracted? and by which kit? and what is ELISA extraction buffer? 

Author Response

L42: ".... uniform conditions as regards protective measures..." 

".... uniform conditions for protective measures..."

Modified

L46-48: "Visual inspection is not always feasible to assess an infection with PLMVd because even if PLMVd can induce a broad variety of symptoms, those are often not stable and disappear with time. "

"PLMVd infection can induce a broad variety of symptoms, that are often unstable and disappear during the season. Visual inspection is therefore not always feasible to assess PLMVd infection."

Modified

L49-51: "The economic impact of PLMVd is mainly characterized by long term problems, like fruits alteration, reduced tree longevity, and increased susceptibility to other biotic and abiotic stresses [1]."

 "The economic impact of PLMVd is mainly characterized by long term effects of infection such as, the alteration of fruit, reduced tree longevity, and increased susceptibility to other biotic and abiotic stresses [1]."

Modified

Also: if what way is the fruit altered? Maybe expand or clarify the statement. Done

L54-57: "In that respect, rapid, effective and low-cost diagnostic tests are highly necessary to allow an early and reliable PLMVd detection. Generally, viroid detection can be tricky and due to the lack of the coat protein there is the consequent impossibility to make use of massive screening test like ELISA. While molecular detection of PLMVd is effective, it  is made more difficult by the intrinsic property to evolve rapidly of PLMVd, quickly accumulating changes in the genome [1,3,4]. " 

"Rapid, effective and low-cost diagnostic tests are highly necessary to allow an early and reliable PLMVd detection. Generally, viroid detection can be tricky and due to the lack of the coat protein there is the consequent impossibility to make use of massive screening test like ELISA. Although molecular detection of PLMVd is effective, the rapidly evolving PLMVd genome accumulates changes that can potentially compromise a sequence-based detection assay [1,3,4]. " Modified

L275: "...variants often escaped to the detection ..."

"...variants often go undetected ..." Modified

L289: "...all the year." 

"...year round." or "...over the whole season" Modified

L295: "...higher by the developed... "

"...higher for the developed... " Modified

L309-312: "Another important consideration is that the result obtained in using the rapid extraction methods is very interesting. In fact, they proved to be reliable enough to be used in massive analysis, thus facilitating a wider, earlier and economic control, preventing the spread of PLMVd in propagation material and in field. "

"Here we provide evidence that rapid extraction methods can provide reliable enough results to be used in routine high-throughput diagnostics, preventing the spread of PLMVd in propagation material and in field. " Modified 

QUESTIONS FOR CLARIFICATION

L97: Please clarify which RNA extract was used for this optimisation? A reference for the protocol for the extraction was added, since the extraction method used was reported more extensively in the materials and methods.

L111: Please provide more information on how the normalisation was performed. The information were reported on a specific paragraph of the Materials and Methods – Statistical analysis; the format of the journal Viruses reports M&M section at the end of the work. We tried to better explain the concept also in results section L112-115 (clear version) in the text.

L167: Why no Zymo TL comparison? Because as reported, the combination of TL+ Zymo is the only procedure already validated and resulted the best protocol in TPS on PLMVd detection organized in 2014 (Luigi and Faggioli 2014). According to that, this protocol was used as reference to normalize the others.

L176: consider starting a new paragraph with a separate subheading. Done

L193-195: Wording can be confusing. Consider re-phrasing. Modified

Figure 7: Please add headings to graphs with extraction method. Done

Figure 8: Why are these two graphs in the same figure? Because both reported DCq values with rapid extraction protocols

  1. a) Leaf vs phloem comparison. Not clear that hose were the two best methods depicted in Fig7. They were mostly comparable, We tried to better clarify in the text.
  2. b) This is a nylon vs paper comparison. But now with Bioreba buffer?  

Somewhere I am missing the link.

In certification programs, most prune viruses are tested by DAS-ELISA on leaves using Bioreba buffer for homogenization. The use of a single homogenization step with the same buffer for serological and molecular testing could speed up the procedures and make them easier. For this reason, in this work we included the homogenization buffer used in DAS-ELISA in the extraction protocols tested. This is explained in the discussion.

L216: Sample? Does this refer to the RNA extracted? and by which kit? and what is ELISA extraction buffer? The list of the samples used in the TPS was reported in Table 2 and 6, we choose two healthy samples, three PLMVd-infected samples and one no-target. From those samples phloem and leaf tissues were collected and prepared according to the indication reported in the results (set A, set B, set C, set D) and better explained in the Materials and Methods paragraph - Test performance study. We tried to reformulate the sentence to better clarify.

Reviewer 2 Report

The manuscript titled "Development and validation of a duplex real time RT-PCR for the detection of peach latent mosaic viroid and comparison of
different nucleic acid extraction protocols" was well written and the detection protocol extensively and rigorously tested. This method will greatly improve diagnostics for this pathogen in many labs.

Author Response

Thank you for appreciating our work

Reviewer 3 Report

The presented work is overall very well designed well written and presented. Some parts, however, need better and clearer presentation and structure: in particular Fig. 3 needs some efforts for both legends and caption text.

There are many other small issues to be corrected:

Lines 41-45: this sentence is too complicated to understand. Please improve the structure and comprehensibility. Suggestion: “In 2019, when the Regulation (EU) 2019/2072 was implemented in Europe, the European Commission established uniform conditions regarding protective measures against pests of plants. Peach latent mosaic viroid (PLMVd) was included in the list of pests for which visual inspection, and, in particular cases, sampling and testing are required for Prunu  persica (Annex IV, part J)”.

Line 52: thaught through or spreads with infected propagation material

Line 57-59: this sentence is not comprehensible, please improve

Line 73: the duplex assay was tested. Please avoid using the same words repeatedly.

Line 73: Tested in vivo? No kidding?

Line 82: analysed in silico

Line 98: of probes reportet above? Where is above?

Line 110/170/391: TRNA / T-RNA – use consistent abrreviations

Lines 115-121: X axis annotation of the figure unclear/unreadable. Highlight the extraction method according to Fig. 6 and chose different colors for the test/primer systems. The caption text should be better structured. This Fig. is actually very difficult to understand.

Lines 169-175: for sake of better comprehensibility: indicate which extraction procedure (benchmark protocol) the Cq were normalized to.

Line 222-224: The caption text should be more comprehensive. What is a set? Why are the results of one specific sample so different between the sets?

Line 384-385: indicated tube and type and number of beads used for homogenizing the samples in the Tissue Lyzer. Maximum speed = 30 Hz (can be indicated!) Was the phloem tissue fresh, frozen, lyophilized?

Line 387: how was it homogenized? Bead beating?

Author Response

The presented work is overall very well designed well written and presented. Some parts, however, need better and clearer presentation and structure: in particular Fig. 3 needs some efforts for both legends and caption text. We reformulated both the legend and the caption in the text.

There are many other small issues to be corrected:

Lines 41-45: this sentence is too complicated to understand. Please improve the structure and comprehensibility. Suggestion: “In 2019, when the Regulation (EU) 2019/2072 was implemented in Europe, the European Commission established uniform conditions regarding protective measures against pests of plants. Peach latent mosaic viroid (PLMVd) was included in the list of pests for which visual inspection, and, in particular cases, sampling and testing are required for Prunu  persica (Annex IV, part J)”. Modified

Line 52: thaught through or spreads with infected propagation material Done

Line 57-59: this sentence is not comprehensible, please improve Done

Line 73: the duplex assay was tested. Please avoid using the same words repeatedly. Done

Line 73: Tested in vivo? No kidding? Modified

Line 82: analysed in silico Modified

Line 98: of probes reportet above? Where is above? Modified

Line 110/170/391: TRNA / T-RNA – use consistent abrreviations We uniformed all using TRNA

Lines 115-121: X axis annotation of the figure unclear/unreadable. Highlight the extraction method according to Fig. 6 and chose different colors for the test/primer systems. The caption text should be better structured. This Fig. is actually very difficult to understand. Modified

Lines 169-175: for sake of better comprehensibility: indicate which extraction procedure (benchmark protocol) the Cq were normalized to. Reformulated

Line 222-224: The caption text should be more comprehensive. What is a set? We better clarify in the text

Why are the results of one specific sample so different between the sets? Because as reported in the paragraphs of the TPS (both in results and Materials and Methods) each sample was prepared in different ways depending on the set. The varying parameters of preparation included matrix (leaf or phloem tissue), extraction or grinding buffer, printing/spotting procedure and type of membrane. The same sample prepared in such different conditions may provide different test results. The aim of the TPS was purposedly to investigate the effect of extraction on test results.

Line 384-385: indicated tube and type and number of beads used for homogenizing the samples in the Tissue Lyzer. Maximum speed = 30 Hz (can be indicated!) Was the phloem tissue fresh, frozen, lyophilized? Modified

Line 387: how was it homogenized? Bead beating? Added

Reviewer 4 Report

This manuscript describes the use of dRT-qPCR for the detection of Peach latent mosaic viroid. A "new"protocol for the detection of this viroid is described. I think this is a methodological paper with low novelty that should be published in a methods-journal. In my opinion it is of only very specific interest.

It is difficult for the reader to know at the beginning what this "new" method really includes.
line 82: explanation of  and  differences between the two published protocols is necessary and how these were fused

fig 1: improve explanation of what can be seen. each curve=one isolate? or one sRT-qPCRs? why are the red curves flatter?

line 252: Explain the deviation in the protocol of lab 3

line 271: What do you nean with "several issues"? several factors?

line 285: If the method should be cheap, why use magnetic beads that are expensive?

Author Response

This manuscript describes the use of dRT-qPCR for the detection of Peach latent mosaic viroid. A "new"protocol for the detection of this viroid is described. I think this is a methodological paper with low novelty that should be published in a methods-journal. In my opinion it is of only very specific interest.

We thank the reviewer to highlight this aspect, but we think that this point has been addressed and overcome by the other  reviewers and by the Editor's comment. Everyone considers the described method as “new”. Additionally, the test here presented could pave the way to the development of similar tests, following the same rationale, for the detection of other pathogens presenting the same difficulties.

It is difficult for the reader to know at the beginning what this "new" method really includes.
line 82: explanation of  and  differences between the two published protocols is necessary and how these were fused

The reviewer rightly asks what the differences are (between this method and the previous ones) and how the previous two methods were merged. All this information is reported in paragraph 4. "Materials and Methods" starting from line 346 up to line 363 and in Table 7. Probably the Reviewer's request is due to the fact that paragraph 2. "Results" precedes paragraph 4. "Materials and Methods". This is due to the format of the journal Viruses and not our choice.

fig 1: improve explanation of what can be seen. each curve=one isolate? or one sRT-qPCRs? why are the red curves flatter? Done

line 252: Explain the deviation in the protocol of lab 3 Added

line 271: What do you nean with "several issues"? several factors? Modified

line 285: If the method should be cheap, why use magnetic beads that are expensive? It was probably not clearly pointed out in the text that the method has the advantage of being cheap when using rapid extraction methods. In fact, in this case neither magnetic beads nor expensive extraction kits are used. In fact, the validation parameters (sensitivity, specificity, accuracy, reproducibility) obtained with the rapid extraction protocols are equivalent or slightly less than the classic ones, thus making the new method adaptable also to cheaper extractions. This statement has been improved within the text.